# Probiotic and *Triticale* Silage Fermentation Potential of *Pediococcus pentosaceus* and *Lactobacillus brevis* and Their Impacts on Pathogenic Bacteria

**DOI:** 10.3390/microorganisms7090318

**Published:** 2019-09-04

**Authors:** Ilavenil Soundharrajan, Dahye Kim, Palaniselvam Kuppusamy, Karanan Muthusamy, Hyun Jeong Lee, Ki Choon Choi

**Affiliations:** 1Grassland and Forage division, National Institute of Animal Science, Rural Development Administration, Cheonan 31000, Korea; 2Center for Research on Environmental Disease, College of Medicine, University of Kentucky, 236 HSRB 1095 VA Drive, Lexington, KY 40536, USA; 3Jangsu Agriculture Technology Center, Jeollabuk-do, Jangsu-gun 55640, Korea

**Keywords:** *Pediococcus pentosaceus*, *Lactobacillus brevis*, silage fermentation, antibacterial, probiotics features

## Abstract

The purpose of this study was to identify potent lactic acid bacteria that could have a great impact on triticale silage fermentation at different moisture levels and determine their anti-bacterial activity and high probiotic potential. For this purpose, *Pediococcus pentosaceus* (TC48) and *Lactobacillus brevis* (TC50) were isolated from fermented triticale silage. The fermentation ability of these isolates in triticale powder was studied by an ensiling method. TC48 had higher ability to ferment silage powder by increasing the lactic acid content of silage than TC50. Extracellular supernatant (ECS) of TC48 and TC50 exhibited strong antibacterial effects (inhibition zone diameters: 18–28 mm) against tested cattle pathogenic bacteria with minimum inhibitory/ minimum bactericidal concentrations (MIC/MBC) values of 5.0–10 mg/mL and 10–20 mg/mL, respectively. Extracellular supernatant (ECS) of TC48 and TC50 showed antibacterial activities on *E. coli*, *P. aeruoginosa*, *S. aureus* and *E. faecalis* through destruction of membrane integrity as confirmed by decreased viability, and increased 260 nm absorbing material in culture filtrate of pathogenic bacteria exposed to ECS of both strains. TC48 and TC50 strains exhibited high tolerance to artificial gastric, duodenal and intestinal fluids. TC48 showed good hydrophobicity and auto-aggregations properties. TC48 and TC50 significantly co-aggregated with *E. coli*, *P. aeruoginosa*, *S. aureus* and *E. faecalis* in a time-dependent manner. In summary, all of the bacteria had a positive impact on at least one functional property of the silage during the fermentation process. However, the addition of *P. pentosaceus* (TC48) and *L. brevis* (TC50) yielded the greatest silage quality improvement, having high antibacterial and probiotic properties.

## 1. Introduction

There are severe concerns about the increasing incidence of bacterial diseases in cattle that affect animal health and productivity [1,2,3]. Bacterial infections are responsible for mastitis and diarrhoea in cattle. *E. coli*, *S. enteritidis*, *S. aureus* and *E. faecalis* are common pathogens that affect cattle health [3,4,5,6,7]. Furthermore, *P. aeruginosa* affects animal health by causing many diseases in both livestock and domestic pet animals [8,9,10,11,12,13].

Infectious pathogens can become resistant to current antimicrobial agents, leading to difficulty in treating diseases that might cause mortality [14]. Failure of antibiotics to prevent infections demands the discovery of alternative agents having new mechanisms of actions without side effects. Products of natural origin can offer a diversity of active ingredients and mechanisms of action. Among natural products, probiotic bacteria play key functions in human and animal health. Probiotics are live microorganisms that, when administered in adequate amounts, confer a health benefit on the host [15,16]. Probiotic lactic acid bacteria (LAB) plays a crucial role in silage production from grasses, legumes and other plants as feed for animals. Forage preservation via ensiling method has become more attention because it provides consistent, reliable and predictable feed supply for animal productions. Unavoidable losses of digestible nutrients caused by plant oxidation, microbial population in plants, proteolytic activity, clostridia fermentation, microbial deamination and decarboxylation of amino acids may negatively affect conservation efficiency increases energy and nutrient losses and cause an accumulation of anti-nutritional compounds in silage samples [17]. An addition of LAB, to forage the sample whilst ensiling enhances the acidification of silage, the main criteria producing a different type of organic acids, which includes lactic acid, acetic acid, butyric acid, succinic acid and others. The acidified environment prevents undesirable microbial growth in the silage such as yeast, molds and Listeria species during fermentation. The spoilage microorganisms not only affect the nutritional quality of silage but also influence animal health and its products. LAB has long been used as inoculums to improve low/high moisture silage quality worldwide and also in improving milk production, increasing body weight and efficiency of feed intake [18,19]. Interestingly, LAB exerts antibacterial and antifungal activities against human and animal pathogens by its secondary metabolites [20,21,22]. Hydrogen peroxide, (H_2_O_2_), carbon dioxide (CO_2_), diacetyl, and bacteriocins produced by LAB can also inhibit undesirable microbial growth [20,23].

Triticale is a breed crop species between the wheat and rye; it was bred first in German laboratories in 19th century [24]. The way in which triticale will be used is based on its characteristics of species. The chemical compositions of most of the species are closer to wheat, used in human and animal nutrition. It is used for the preparation of concentrated animal feed for both ruminants and non-ruminants due to its higher total protein compared to starch. [25,26,27]. Currently, triticale is used as a diet for domestic and farm animals. It can be replaced usages of barley and other cereals. The advantage of triticale over other is higher biomass, fast growth in spring and a longer mowing time [28]. The purpose of the present study was to identify and characterize efficient probiotic lactic acid bacteria (LAB) and investigate their effects on triticale silage fermentation, and cattle pathogenic bacterial growth. 

## 2. Materials and Methods

### 2.1. Sample Collection and Isolation of Lactic Acid Bacteria

Triticale grass was collected from Jangsoo, South Korea. The samples were brought aseptically to the laboratory for isolation of lactic acid bacteria (LAB). One gram of the sample was dissolved in 10 mL distilled water and kept in a rotary shaker at 150 rpm for 30 min. After that, tenfold serial dilution was performed. Serially diluted samples were spread on Man Rogosa and Sharpe agar medium (MRS agar; CONDA, Madrid, Spain). Plates were incubated at 30 °C for 72 h, observed bacteria growth and then purified on MRS agar. LAB colonies were further streaked on bromo cresol purple agar medium (BCP), this being a selective medium to confirm the identity of LAB strains. All strains were maintained on MRS agar medium at 4 °C for short time storage [22].

### 2.2. Inoculums Preparation for Silage Production 

The selected strains were cultured in MRS broth (CONDA, Madrid, Spain) for 24 h and then centrifuged at 4000×·*g* for 30 min at 4 °C. The cell-free supernatant was removed and pellets were washed twice with phosphate buffered saline (PBS, pH 7.4) and diluted in the same buffer. The bacteria colonies were enumerated by Quantom Tx Microbial cell counter with Quantom total cell staining kit (Logos biosystem, Gyeonggi-do, South Korea). In brief, one microliter quantum total cell staining, one microliter total cell staining enhancer and ten mirocliters diluted samples were mixed well and kept at room temperature for 30 min. Eight microliters of cell loading buffer were then added to mixtures and mixed thoroughly without bubbles. Six microliters of prepared sample were loaded onto quantom M50 cell counting slide and centrifuged at 300×·*g* for 10 min and then bacteria were counted with a quantom with the light intensity level set to 5 for most bacterial cells

Triticale powder (heading stage) was obtained from Jangsoo, South Korea and prepared samples with different moisture contents (40%, 50% 60%, and 70%) using sterile distilled water under sterilized condition. A hundred gram of each sample at different moisture levels were taken in polythene bag size 28·× 36 cm (Aostar Co., Ltd., Seoul, Korea) and categorized into four groups, each consist of three replicates. Group I—control without inoculums; Group II S—control (sterilized with ethylene oxide gas), Group III samples were treated with LAB at different moistures; Group IV samples treated with standard lactic acid bacteria (KACC91961P). 

Group I and group II samples were treated with PBS alone; Group III were treated with isolated lactic acid bacteria (1·×·10^5^), Group IV samples treated with company lactic acid bacteria (KACC91961P) and the air was evacuated from the bags using a vacuum sealer (Food saver V48802, MK corporation, Korea). A total of 210 bags were prepared and stored for 30 days in the laboratory at room temperature (range, 25–28 °C). After fermentation, the bags were opened for further analysis.

#### 2.2.1. Fermentative Profile 

Water extracts of silage samples were prepared by weighing 20 g of silage in 80 mL deionized water and kept in shaking incubator at 150× *g* (Vision Scientific Co Ltd., Daejeon-Si, Korea) for 30 min at 4 °C. Then all samples were centrifuged at 4000× *g* at 4 °C for 20 min. The aqueous extract was divided into two portions: One portion was used to measure the pH using a combination electrode (inolab pH meter, Thomas Scientific, NJ, USA) and determine the concentrations of lactic acid, acetic acid and butyric acid using HPLC (HP1100 Agilent Co, Santa Clara, CA, USA) as described previously [19].

#### 2.2.2. Quantification of Microbial Populations 

The second portion of aqueous extracts was used to quantify the LAB, Yeast and fungi population. The bacteria colonies were enumerated by Quantom Tx Microbial cell counter with Quantom total cell staining kit as described early (Logos biosystem, Gyeonggi-do, Korea). Cultivations of fungi and yeast were performed in potato dextrose agar medium (Fisher Scientific, Frederick, MD, USA) and petrifilm (Thermofisher Scientific, Coon Rapids, MN, USA), respectively.

#### 2.2.3. Extract Preparation from Fermented Triticale Silage for Anti-Bacterial Study

All three replicates of fermented powder from each group were soaked in 70% methanol and kept in shaking incubator for 24 h at 150× *g*. The extract was filtered using filter cloth followed by Whatman No. 4 and Whatman No. 2 filter papers. The solvent in the extract was evaporated with a rotary vacuum evaporator (Buchi Rotavapor R-114, Marshall Scientific, Hampton, UK). The concentrated extract was then freeze-dried, lyophilized by freeze dryer (Conditions: 50 T and pressure at −80 °C; Ilshin Biobase, Gyeonggi-do, Korea). Lyophilized pellets were dissolved in sterile water 10 mg/mL) and used for the antibacterial study. 

#### 2.2.4. Characterization of Isolates by Biochemical and Molecular Methods

Potent strains were selected based on above said experiments and analyzed their biochemical properties and enzyme productions using API 50 CH and API-ZYM kits, respectively (Marcy l’Etoile, France). Antibiotic sensitivity was tested with disc diffusion method as described previously [29]. 16S rRNA gene sequencing was performed at Solgent Co. (Seoul, South Korea) using the Sanger method [30]. Genomic DNA was isolated and purified using a QIAquick^®^ kit (Qiagen Ltd., Crawley, UK). Amplicons were sequenced using universal primers 27 F (5′ AGA GTT TGA TCG TGG CTC AG 3′) and 1492 R (3′ GCT TAC CTT GTT ACG ACT T 5′). The aligned KCC-29 16S rRNA sequence was subjected to a BLAST search ((GenBank non-redundant database (NCBI)). The 16S rRNA sequence obtained from TC48 (KCC-45) and TC50 (KCC-44) isolates were deposited to the GenBank (Accession Nos.: MN049503 and MN049504).

### 2.3. Intracellular Supernatant (ICS) and Extracellular Supernatant (ECS) Preparations 

TC48 and TC50 were inoculated in the MRS broth in 500 mL conical flasks and incubated at 37 °C for 48 h in shaking incubator at 150× *g* (Vision Scientific Co Ltd., Daejeon-Si, Korea). After that, the broth was centrifuged at 4 °C for 30 min at 4000×·*g*. Bottom LAB pellets were suspended in PBS containing protease inhibitors and lysed by speed Mill PLUS with beads (Analytik Jena, Jena, Germany). Supernatants were then collected after centrifugation. All supernatants were filtered and lyophilized by freeze dryer (Conditions: 50 T and pressure at −80 °C; Ilshin Biobase, Gyeonggi-do, Korea). These lyophilized pellets were dissolved in PBS to have a concentration of 20 mg/mL for antibacterial study. 

### 2.4. Antibacterial Activity-Well Diffusion, MIC/MBC, and Time Killing Assay Methods

Pathogens such as *P. Aeruoginosa* (KACC-10185)*, E. faecalis* (KACC-11304)*, E. coli* (KACC-1005)*, and S. aureus* (KACC-10768) were obtained from the Korean Agricultural Culture Collection (KACC), South Korea. For well diffusion assay, 100 μL of 24 h fresh pathogenic bacteria such as *P. Aeruoginosa, E. faecalis*, *E. coli and S. aureus* were spread onto nutrient agar plates. Wells were made on the agar using a well cutter. After 100 µL of the prepared extract was loaded into each well and incubated at 37 °C for 48 h, inhibitory zone (mm) was then monitored and calculated. Minimum inhibitory concentration (MIC) and minimum bactericidal concentration (MBC) of extracts were analyzed according to the method described by [31] with slight modifications. Briefly, two-fold serial dilutions of ECS of LAB were prepared in nutrient broth in 96-well plates. A 0.5 McFarland standardized bacterial inoculum was added to each well. Plates were then incubated at 37 °C for 24 h. Control well and media were also included. MIC and MBC were then analyzed using a by Quantom Tx Microbial cell counter with Quantom viable cell staining kit (Logos biosystem, Gyeonggi-do, South Korea). According to [31,32], time-killing assay was performed with some modifications. Fresh bacterial cultures were suspended in nutrient broth and adjusted to an optical density of approximately 10^6^·CFU/mL. Different concentrations of ECS (1.25 to 20 mg/mL) of TC48 and TC50 were added to the inoculum suspension and incubated at 37 °C. Aliquots were removed from the experimental well after 12, 24, 36, and 48 h of incubation and measured optical density at 600 nm. Impacts of ECS of TC48 and TC50 on releases of 260-nm-absorbing materials were also analyzed [33,34].

### 2.5. Probiotic Characterization of LAB

#### 2.5.1. Preparations of Juices for Simulated Gastrointestinal Tract Tolerance Test 

The pH of PBS was adjusted to 2.5, 5.0, and 8.0 using 1 M HCl or 1 M NaOH followed by autoclaving at 121 °C for 15 min. Simulated gastric juice (GJ) was prepared by mixing PBS (pH 2.5) with 3 mg/mL pepsin (St. Louis, USA) followed by filtering with 0.2 μM membrane filter (Acrodisc, Pall Corporation, NY, USA). Duodenal juice (DJ) was prepared by adding 0.3% bile salt (Sigma-Aldrich St. Louis, USA) and 0.1% trypsin to PBS (pH 5). Intestinal juice (IJ) was prepared by supplementing PBS (pH 8) with 0.1% trypsin (Gibco- Thermo Fisher Scientific, St. Louis, MO, USA) followed by filtering with 0.2 μM membrane filter. 

#### 2.5.2. Survival Ability of LAB in Gastric Juice, Duodenal Juice, and Intestinal Juice

Gastrointestinal tract tolerance of TC48 and TC53 was analyzed with slight modifications [35]. Briefly, TC48 and TC53 were grown in MRS broth at 37 °C for 24 h. Pellets were harvested by centrifugation at 4000× *g* for 15 min at 4 °C. These cells were then washed with PBS (pH 7.4) twice, resuspended in PBS, and counted using a quantum live cell staining kit. Resuspended (1·×·10^6^) bacteria were added to simulate GJ and incubated at 37 °C for 3 h. After that, 1 mL of cell suspension was transferred from GJ into 9 mL of simulated DJ and incubated at 37 °C for 3 h. After the incubation, 1 mL of cell suspension was transferred from DJ into 9 mL of simulated IJ and incubated at 37 °C for 3 h. Before and after 3 h of incubation, TC48 and TC50 in each juice were analyzed with a Quantom Tx Microbial cell counter with Quantom viable cell staining kit (Logos biosystem, Gyeonggi-do, South Korea).

### 2.6. Hydrophobicity of TC48 and TC50 

Microbial adhesion to hydrocarbons determines the ability of bacteria to adhere to intestinal walls. Hydrophobicity of bacteria in the presence of chloroform and xylene as hydrocarbons was determined [36].

### 2.7. Hemolytic Activity of TC48 and TC50

The hemolytic activity was evaluated [37]. The *E. coli* used as a control for haemolysis.

### 2.8. Autoaggregation and Coaggregation Assay

Aggregation of cell membranes with interacting surfaces was examined [38]. Briefly, bacterial cells were harvested by centrifugation at 5000×·*g* for 15 min at 4 °C. Bacterial pellets were then washed twice with phosphate-buffered saline solution (pH 7.2), re-suspended in phosphate buffer, and incubated at 37 °C for 3 h. Aliquots (100 µL) were withdrawn at intervals of 1 h from the upper layer and mixed with 3.9 mL of phosphate buffer saline. Absorbance was read at a wavelength of 600 nm. Autoaggregation was calculated as 1 − (A_t_/A_0_)·×·100, where A_t_ was the absorbance at time A_t_ = 1, 2, and 3 and A_0_ was absorbance at t = 0. For co-aggregation, sample preparation was that same as that for autoaggregation assay. Equal volumes (LAB and pathogenic bacteria) of cell suspensions were mixed by vortexing for 10 s. Absorbance were then read at a 600 nm after 3 h of incubation at room temperature. Control tubes were also included at the same time individually [39]. Coaggregation was calculated as ((A*x·*+·A*y*)/2) − A(*x·*+*·y*) A*x·*+*·*A*y*)/2*·*×*·*100, where *x* and *y* were optical densities for the two strains individually and *x·*+*·y* was the optical density of mixed strains.

### 2.9. Statistical Analyses

All numerical data were obtained from three independent experiments. Results are presented as means ± STD. Significant differences were analyzed with SPSS16 software using analysis of variance (one-way ANOVA, multivariate analysis, included post hoc, Duncan and descriptive analysis parameters) followed by the least significance difference test. *p* values of less than 0.05 were considered as statistically significant. 

## 3. Results

### 3.1. Isolation and Selection of Potent Strains Based on Antimicrobial Activity and Fermentation Improvements in Triticale Silage 

In the current study, we isolated several lactic acid bacteria (LAB) from fermented triticale silage. Preliminarily study indicated that 58 LAB (hereafter named as TC1-TC58) showed significant antibacterial activities against pathogenic *E. coli* and *S. aureus* (data not shown). In addition, all strains in this study were able to reduce the pH of silage and increased fermentation quality of triticale silage at different moisture conditions ranging from 40% to 70% (Appendix A). Among these 58 strains, TC2, TC10, TC16, TC48 and, TC50 displayed adequate growth and enhanced acidification of experimental silages than the control silages (Table 1). 

Mainly, TC48 and TC50 produced a higher amount of lactic acid (*p <* 0.05) with a marginal amount of acetic acid and butyric acid during fermentation than TC2, TC10, TC16 (Figure 1). Furthermore, we analyzed anti-bacterial activities of extracts of fermented silage of triticale. Results showed that the extracts of silage fermented with TC48 and TC50 exhibited higher antibacterial activities against tested bacteria than the control silage. Especially, extract from 60% and 70% moisture showed higher antibacterial activity than extract prepared from other moisture (Table 2). Overall, TC48 and TC50 exhibited potential fermentation ability with significant antibacterial activity than the other isolates. Therefore, we selected these strains for further antibacterial study against cattle pathogenic bacteria and their probiotic potential. The basic physio-chemical results exhibited that the TC48 and TC50 are the gram-positive, non-motile, catalase-negative and mesophilic nature. The shape of the TC48 and 50 were circular and rod shapes, respectively. 16srRNA sequence results revealed that TC48 has corresponded to the *p. pentosaceus* and TC50 has belonged to the *L brevis.*


These strains strongly ferment the different types of sugars such as d-Glucose, d-Fructose, d-Mannose, l-Sorbose, d-Maltose, d-Lactose, Inulin and d-Fucose etc. (Table 3). Besides, TC48 and TC50 produced industrially important extracellular enzymes which include different proteolytic, lipolytic and glycolytic enzymes (Table 4). Antibiotic sensitivity of TC48 and TC50 was screened and it was categorized as moderate susceptibility (M), strong susceptibility (S) and resistant (R). These strains exhibited high sensitivity against commonly used antibiotics (Table 5). TC48 and TC50 did not exhibit hemolytic activity.

### 3.2. Effect of Fresh and Denatured Extracellular Supernatant (ECS) and Intracellular Supernatant (ICS) on Cattle Pathogenic Bacteria

Since our results showed that TC48 and TC50 had the potential ability to ferment triticale silage at all moisture conditions and controlled undesirable microorganism growth. We next investigated antibacterial activities of ECS, and ICS of TC48 and TC50 against cattle pathogenic bacteria including *P. aeruoginosa, E. faecalis, E. coli and S. aureus.* Fresh ECS of TC48 and TC50 showed strong inhibitory effects against all tested pathogenic bacteria. TC48 showed the highest zone of inhibition against *E. coli* and *E. faecalis* than *P. aeruoginosa* and *S. aureus* whereas TC50 showed maximum antibacterial activity against *E. coli, E. faecalis* and *P. aeruoginosa.* On the other hand, ICS of both bacteria showed much weaker antibacterial activities against tested bacteria than ECS (Table 6). 

Furthermore, we analyzed the effect of denatured ECS and ICS at 90 °C for 1 hrs on selected pathogenic bacteria. When exposing of ECS and ICS to high temperature might have denatured/ inactivated the peptides and proteins produced in MRS broth by the TC48 and TC50. Likewise, heat treatment reduced antibacterial activities of both TC48 and TC50 against all tested pathogenic bacteria. However, after denaturation also exhibited antibacterial against *P. aeruoginosa, E. faecalis, E.coli and S. aureus,* it demonstrated that something other than proteins and peptide also involved in antibacterial activity, because these strains were produced a higher amount of lactic acid with a marginal level of acetic acid and butyric acid in MRS broth (Figure 2). It is also a reason behind the antibacterial activity of ECS after denaturation. Fresh/denatured ECS of all strains showed greater activity against all tested pathogenic bacteria than the fresh/denatured ICS (Table 6).

### 3.3. Minimum Inhibitory Concentration(MIC) and Minimum Bactericidal Concentration (MBC)

We then determined MIC and MBC of ECS (20 mg/mL to 1.25 mg/mL, two fold serial dilutions) of TC48 and TC50 against *P. aeruoginosa, E. faecalis, E. coli and S. aureus*. TC48 showed MIC of 5 mg/mL and MBC of 10 mg/mL against *E. faecalis,* and *E. coli,* whereas MIC and MBC against *P. aeruoginosa,* and *S. aureus* were 10 and 20 mg/mL, respectively. TC50 exhibited MIC of 5.0 mg/ mL and MBC of 10 mg/mL, against all tested bacteria except *S. aureus* (Table 7). 

### 3.4. Time-Dependent Killing of Pathogenic Bacteria by ECS Treatment 

*P. aeruoginosa, E. faecalis, E. coli* and *S. aureus* were treated with different concentration (20–1.25 mg/mL) of ECS of TC48 and TC50 and incubated for 48 h. Growth of pathogenic bacteria was then analyzed at different time points (every 12 h). ECS treatment at a concentration of 20 mg/mL completely arrested the growth of *P. aeruoginosa, E. faecalis, E. coli and S. aureus* throughout the incubation periods (Figure 3 and Figure 4). However, bacterial growth was increased when the concentration of ECS was reduced, although their growth was significantly (*p* < 0.05) less compared to that of the control group without ECS treatment (*p* < 0.05).

### 3.5. Effect of ECS on Releases of 260 nm Materials 

Analysis of 260 nm absorbing material (nucleic acids) from experimental cells treated with specific antimicrobial agents might be an indicator of bacterial cell membrane damage. Thus we analyzed impacts of ECS of TC48 and TC50 on the release of 260 nm absorbing material from *P. aeruoginosa, E. faecalis, E. coli and S. aureus* treated with ECS at MIC. Results showed that to all tested bacteria had rapid increases in the release of 260 nm absorbing material onto peptone water in a time-dependent manner, whereas no changes were noted in control of tested pathogens. The maximum release of 260 nm absorbing material was noted with ECS of TC48 and TC50 at 60 min incubation (Figure 5). 

### 3.6. Survival Rates of LAB in Simulated Gastrointestinal (GI) Tract Conditions

Since LAB showed good anti-bacterial activity against cattle pathogenic bacteria, we next determined whether these LAB could survive or not in simulated GI tract conditions, for the selection/development of strong probiotic strains. We observed general decreases in survival rates of all strains in simulated gastrointestinal juice (GJ), duodenal juice (DJ) and intestinal juice (IJ) after 3 h incubation. Both TC48 and TC50 had survival ability in the harsh condition of GJ, DJ, and IJ (Figure 6A–C).

### 3.7. Hydrophobicity Properties of LAB

Hydrophobicity of LAB was determined using hydrocarbons chloroform and xylene. Percentages of hydrophobicity in the presence of xylene and chloroform for TC48 were 39.22 ± 7.8% and 27.10 ± 1.4% and TC50 were 45.73 ± 6.5% and 28.94 ± 7.5% at 60 min incubation (Figure 6D). 

### 3.8. Aggregations Properties of LAB

For auto-aggregation, TC50 showed a higher auto-aggregation property (%), aggregating immediately and formed a precipitate in the bottom, resulting in a clear solution. However, TC48 did not autoaggregate immediately. Its suspension showed both precipitate and constant turbidity. TC48 required more incubation time for aggregation than TC50 (Figure 6E) In a co-aggregations study, both TC48 and TC50 strongly co-aggregates with *E. coli* than *P. aeruoginosa, E. fecalies, S. aureus*. At the same time, TC50 showed higher co-aggregation ability with *E. coli* and *P. aeruoginosa* (Figure 6F).

## 4. Discussion

Frequent use of different antibiotics in animal husbandry, humans, agriculture and aquaculture, had contributed to the development of antimicrobial resistance [40,41]. Infectious diseases are causing serious concerns for animal health and its productivity [1,2,3]. Silage is the fermented form of animal feed which gives required nutrients to the livestock animals than the fresh/non-fermented forages. It is a very complex fermentation process that shows variability in natural microbial counts, chemical composition, and nutrients availability such as water-soluble carbohydrates and other nitrogenous constituents for microbial growth [42]. The occurrence of desired silage fermentation is closely related to the amount and types of microorganism present in the plant and the water soluble carbohydrates (WSC) level of the silage [43]. Our current goal was to make the good quality silage with high antibacterial efficiency using LAB as additives for livestock’s animals because lactic acid bacteria (LAB) has attracted much attention as biotherapeutic and immunoprophylaxis agents due to the generally recognized as safe (GRAS) status [44]. Recently LAB can be considered as important additives to preserve the silage for a long time and improving its quality via increasing lactic acid production. Also, the addition of LAB prevents the undesirable microbial growth in silage by reducing the pH of the surrounding places [18,19]. 

Given the above said aspects, we isolated and characterized new lactic acid bacteria (LAB) from triticale plant and analyzed their potential roles in silage production at different moisture conditions in-vitro. The faster increases in LAB counts observed in inoculated silages indicated that LAB strains were competitive among the epiphytic communities. The deterioration of enterobacteria and the development of LAB dominants indicate successful fermentation of silages. The speed of the microbial growth closely correlates with the rate of pH decline and production of lactic acid [45]. The reduction of pH of silages is related to the conversion of ensiled materials. The fast initial acidification of promotes a decrease in the enzyme mediated proteolytic activity of the plant itself and prevent the enterobacteria and clostridia growth, which grow until an inhibitory concentration of non-dissociated acids and sufficiently low pH are reached [46].

In our study, acidification induced by natural LAB fermentation in control silages remains a higher pH. The highest pH values in control silages reflected the lower population of LAB and its low efficiency in initiating fermentation and preventing undesirable microbial growth compared to LAB inoculated silages. Lactic acid, acetic acid, and butyric acid are the main acids in silages [47]. In general, lactic acid was found at the highest concentration on silages during the ensiling process and it contributed the most of the decrease in pH during fermentation because it is approximately 10–12 times stronger than other major acids [48].

Here we noted lower lactic acid level in control silages which reflects the lower LAB population and their ability to dominate the fermentation. In general, homofermentative bacteria treatment rapidly decreased pH of silages than the non-inoculated silages [49]. It’s concurrent with our current study, that lower LAB counts were observed in the TC48, homofermentative bacteria inoculated silages than TC50, a hetero-fermentative bacteria treated silages, but higher lactic acid values were produced in TC48 treated silages that indicate the competence of nutrients utilization and the persistence of acidification of the strain. However, both TC48 and TC50 silages had higher lactic acid production than the other inoculated silages.

Furthermore, fermented silages with LAB showed higher antibacterial activities against a panel of cattle pathogenic bacteria based on inhibitory zones with the agar diffusion method. LAB exhibits significant antimicrobial activities against different pathogenic bacteria [29,50,51]. As we discussed earlier, TC48 and TC50 effectively ferment the triticale silage in both low and high moisture level by increasing lactic acid and reducing pH of silage. Extract of fermented silages with TC48 and TC50 exhibits potent antibacterial activities against tested bacteria than the untreated silages extract, especially extract from 60% and 70% moistures fermented silages exhibited strong antibacterial activity than the control and other moistures silage. This antimicrobial activity might be due to TC48 and TC50 produced a higher amount of lactic acid with the marginal level of acetic acid and butyric acid in 60% and 70% moisture silages than the control and other moistures. 

Extracellular cellular metabolites of TC48 and TC50 strains showed antibacterial activities against *E. coli*, *S. aureus*, *P. aeruginosa* and *E. faecalis* based on inhibitory zones with agar diffusion method, MIC/MBC, time killing assay and nucleic acid releases. Exposure of ECS of TC48 and TC50 either completely inhibits or reduces the *E. coli*, *S. aureus*, *P. aeruginosa* and *E. faecalis* viable cell counts. As we discussed earlier, the organic acids and others metabolites from LAB plays the main role in the control of pathogenic bacteria growth, as similarly TC48 and TC50 produced the higher amount of lactic acid with less contents of acetic acid and butyric acid in MRS broth. Even though after heat degradation of ECS, exhibited strong antibacterial activity indicated that other than proteins and peptides, it also involves antibacterial activity. Previous reports also confirmed that the LAB showed potent inhibition activity against *E. coli*, *S. enteritidis* and *S. aureus* [32,52]. Next we analyzed nucleic acid releases from untreated and ECS treated pathogenic bacteria for confirming their cellular integrity status. The plasma membrane of bacterial plays an important barrier function including iron transport, which involves in many functions in cell membrane and in essential enzymatic activities. An elevated level of iron leakage can lead bacterial cell membrane disruption, so it’s necessary to cellular integrity of bacteria for maintain the equilibrium of essential irons in order to maintain energy status and survival [53]. In the current study, ECS of TC48 and TC50 at MIC exhibited deteriorating effects on the release of 260nm absorbing materials from pathogenic bacteria which confirmed that the TC48 and TC50 inhibited the growth of pathogenic microorganisms through disruption of bacterial cellular integrity.

It is well established that the gastrointestinal tract of animal is a favorable environment for low pH and high concentration of bile salts and survival rate of ingested bacteria in the gastrointestinal tract (GIT) is important for selection and development of probiotics as well as understanding the possible mechanisms underlying probiotic functions of beneficial microorganisms [54]. In the present study, the survival rates of TC48 and TC50 were determined and the results were indicated that significant numbers of TC48 and TC50 survived under gastric, duodenal and intestinal juices. The low pH and pepsin have influenced the survival of lactic acid bacteria in simulated gastric juice. We observed general decreases in LAB colonies in all juices. This is due to decreases in ATP production in the absence of glucose in simulated juices. [55]. Glucose provides the ATP pool required for permitting optimal H^+^ extrusion by F_0_–F_1_ ATPase. The function of F_0_–F_1_ ATPase is closely associated with survival rates of LAB by increasing the intracellular pH at a low level in extracellular [56]. Furthermore, TC48 and TC50 survival rates in duodenal juice are related to bile salt because of bile salts provide the detrimental effects to the bacteria [35]. However, compared with gastric juice, the survival rate of TC48 and TC50 was slightly increased in duodenal juice because these strains get more tolerant properties from the previous stress (from gastric juice and duodenal) and it has survival ability in subsequent unfavorable conditions. It is called a cross-protective stress response [57] (Buriti et al., 2010). Bile salts stimulate the expression of genes which are responsible for maintaining the integrity of cell wall and their membranes [58]. The loss of TC48 and TC50 survival rate was decreased in intestinal juice is associated with increases in rising pH (8). It indicates that LAB gets bile tolerance property at the genomic level which protects the cells from the external stress. TC48 and TC50 were considered as most tolerant bacteria against the simulated GIT conditions. 

The adhesion of bacteria is used to determine the hydrophobicity of the cells surface when the bacteria having higher hydrophobicity are expected to have a potent interaction with mucous or epithelial cells. In the present study, TC48 and TC50 showed higher hydrophobicity in the presence of xylene and chloroform. It indicated that TC48 and TC50 could adhere and colonize in the gut [59,60]. Auto-aggregation is an essential factor for biofilm formation that will help with intestinal colonization and bind with intestinal epithelium which controls pathogen adhesion. In our study, TC48 and TC50 exhibited the maximum % of auto-aggregation at 3 h, which indicates the clumping of cells. Further, we analyzed co-aggregation property with *E. coli*, *S. aureus*, *P. aeruginosa* and *E. faecalis*; it revealed that TC48 and TC50 highly co-aggregate with *E. coli* than other tested pathogenic bacteria. TC48 mores strongly co-aggregate with *E. coli* only, while TC50 is highly co-aggregate with *E. coli* and *P. aeruginosa*. It indicated that both TC48 and TC50 might have the potential to colonize the intestine. The co-aggregation of probiotics with pathogenic bacteria will kill them by its action of antimicrobial agents produced from probiotics [61].

## 5. Conclusions

All of the new strains tested had a positive effect on triticale silage fermentation at different moisture conditions. However, the addition of *P. pentosaceus* (TC48) and *L. brevis* (TC50) had a positive impact on all of the tested parameters and changed silages characteristics; especially, the strains enhanced the production of lactic acid with a marginal amount of acetic acid and butyric acid. In addition, adding these strains decreased the pH of silage, which deteriorated undesirable microbial growth during fermentation and showed potent antibacterial activity with higher probiotics properties, which resulted in a better silage quality that improved the commercial inoculants. Therefore, we suggest that these strains have the potential for use as silage inoculants, but it must be tested in whole plant silage development with different forages. This result obtained at the laboratory level experiment and must be needed to confirm clostridial, and enterobacteria growth before/after inoculants treatment in experimental silages under field conditions. 

## Figures and Tables

**Figure 1 microorganisms-07-00318-f001:**
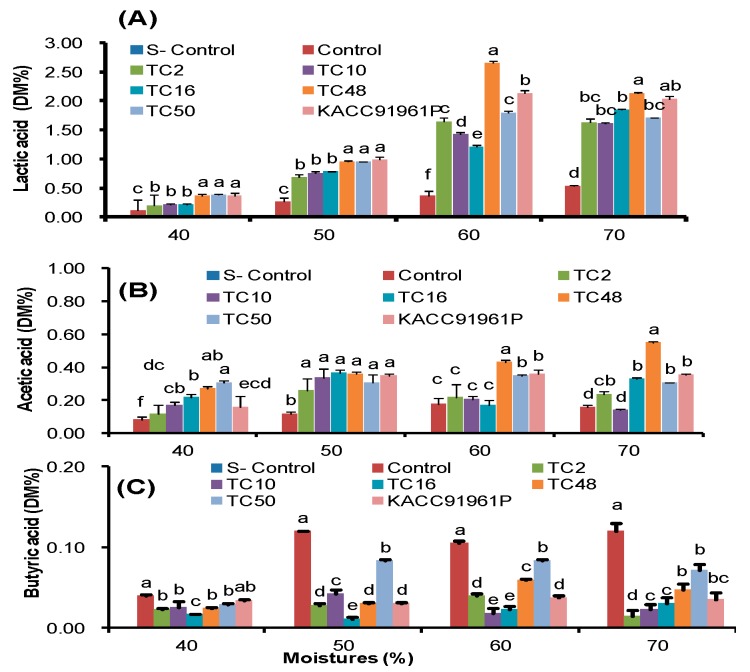
Fermentative metabolites profile of experimental silages at different moistures.Percentage of lactic acid (**A**), acetic acid (**B**) and butyric acid (**C**) in fermented silages quantified by HPLC.S - Control: Sterilized with ethylene oxide gas, DM: Dry matter. Data were represented as mean of standard deviation (mean ± STD, *n* = 5); a, b, c, d, e and f Means with different letters within a column are significantly different at the 5% level.

**Figure 2 microorganisms-07-00318-f002:**
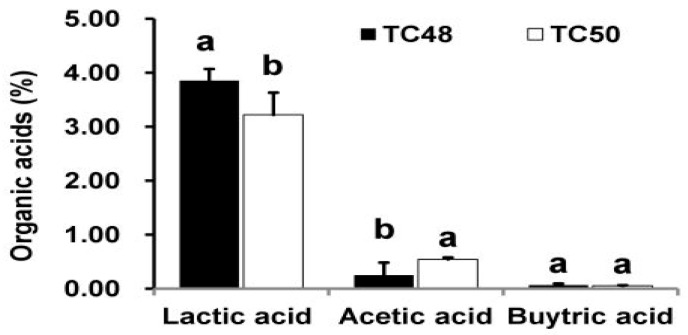
Organic acids production in Man Rogosa and Sharpe agar medium (MRS) broth. Percentage of lactic acid (**A**), acetic acid (**B**) and butyric acid (**C**) in MRS broth quantified by HPLC. a and b means with different letters within a column are significantly different at the 5% level.

**Figure 3 microorganisms-07-00318-f003:**
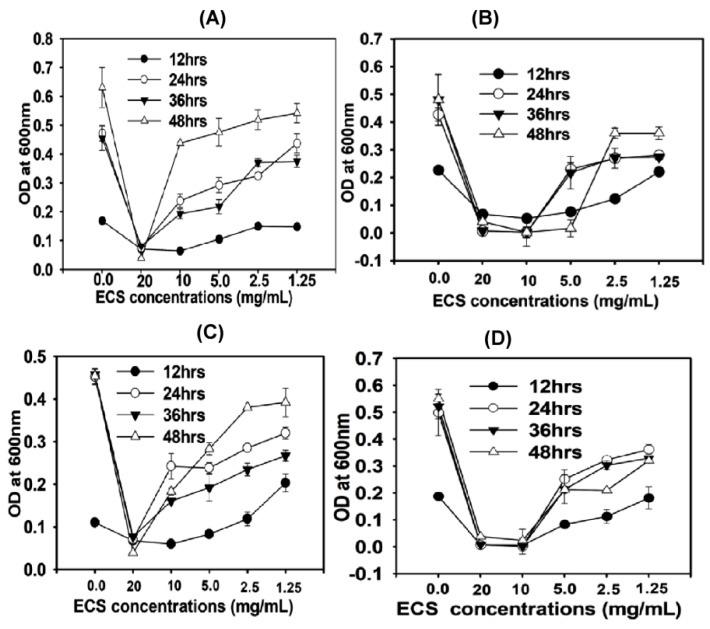
Antibacterial effects of ECS of TC48 on pathogenic bacteria as measured by time killing assay method. ECS of TC48 at different concentrations ranges from 20–1.25 mg/mL on *E. coli* (**A**), *E. faecalis* (**B**), *P. aeruginosa* (**C**) and *S. aureus* (**D**). Data represent the mean ± STD, (*n* = 5).

**Figure 4 microorganisms-07-00318-f004:**
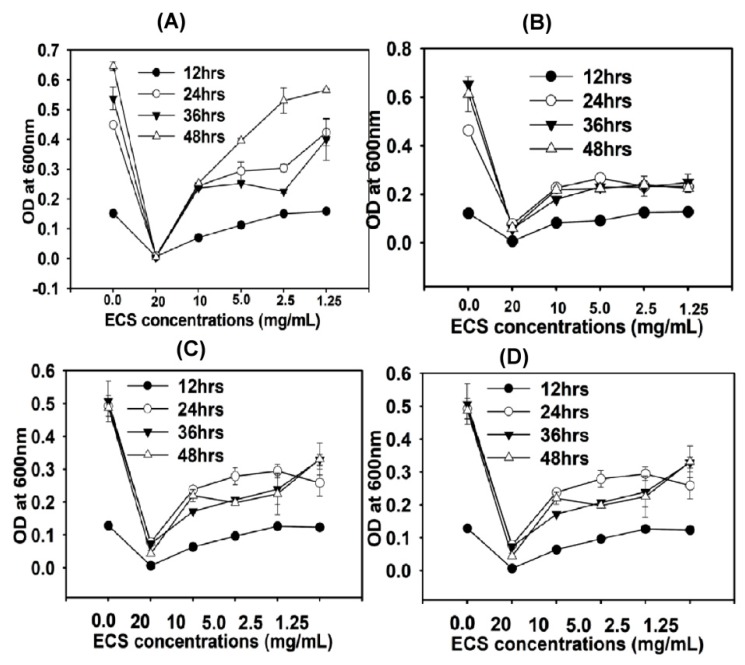
Antibacterial effects of ECS of TC50 on pathogenic bacteria as measured by time killing assay method. ECS of TC48 at different concentrations ranges from 20–1.25 mg/mL on *E. coli* (**A**), *E. faecalis* (**B**), *P. aeruginosa* (**C**) and *S. aureus* (**D**). Data represent the mean ± STD, (*n* = 5).

**Figure 5 microorganisms-07-00318-f005:**
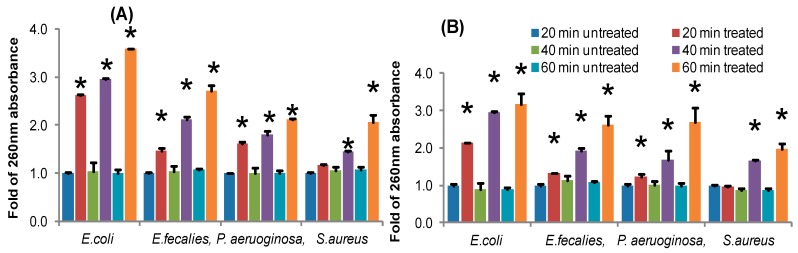
Effect of TC48 and TC50 on the release rate of 260 nm absorbing materials from pathogenic bacteria. (**A**) Fold of 260 nm absorbing materials releases by ECS of TC48 treated pathogenic bacteria; (**B**) fold of 260 nm absorbing materials releases by ECS of TC50 treated pathogenic bacteria. Data represent the mean ± STD (*n* = 5), * Significance between treated and untreated groups at *p* < 0.05 level.

**Figure 6 microorganisms-07-00318-f006:**
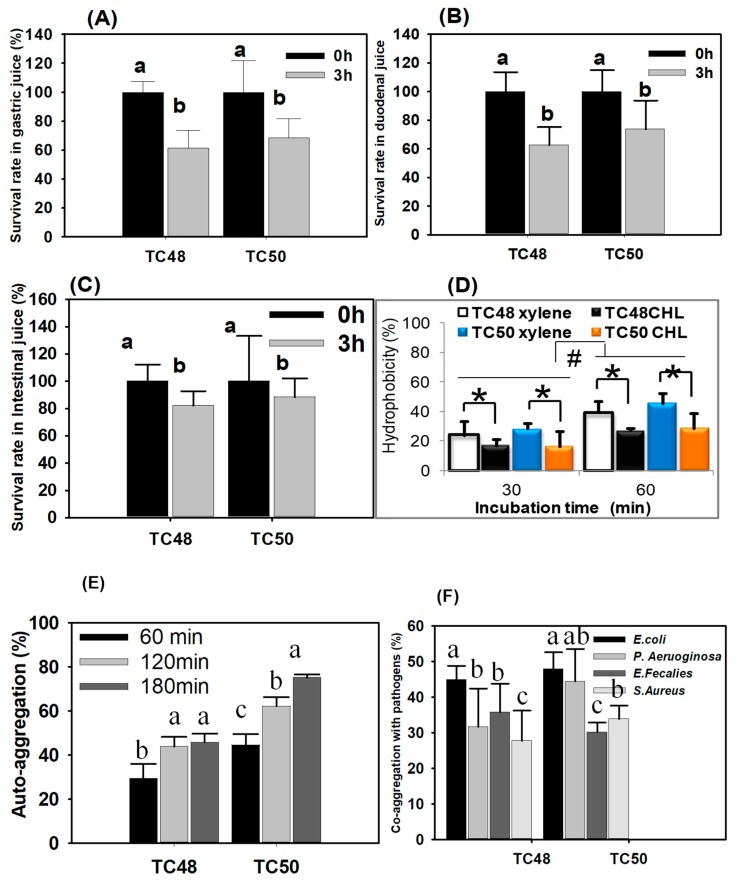
Probiotics feature of TC48 and TC48—in vitro. (**A**) Survival rate of TC48 and TC50 in gastric juice (GJ); (**B**) survival rate of TC48 and TC50 in duodenal juice (DJ); (**C**) survival rate of TC48 and TC50 in intestinal juice (IJ); (**D**) percentages of hydrophobicity for TC48 and TC50; (**E**) percentages of auto-aggregation for TC48 and TC50; (**F**) percentages of co-aggregation for TC48 and TC50 with pathogenic bacteria. Data represent the mean ± STD (*n* = 5, *p* values 0.05). * *p* < 0.05 significance between xylene and chloroform (CHL) ^#^
*p* < 0.05 indicates significance between 30 min and 60 min. Different letters a, b, and c within a column indicate significant differences between time; between pathogenic bacteria.

**Table 1 microorganisms-07-00318-t001:** Average microbial population and pH of triticale silages as a function of the microbial inoculants after fermentation.

Groups	^1^ Log10(CFU g^−1^)				^2^ pH			
	Moistures (%)				Moistures (%)			

	40	50	60	70	40	50	60	70

S–Control	0.00	0.00	0.00	0.00	6.5 ± 0.12 ^a^	6.7 ± 0.16 ^a^	6.7 ± 0.43 ^a^	6.6 ± 0.17 ^a^
Control	6.2 ± 0.18 ^a^	8.4 ± 0.02 ^a^	8.5 ± 0.01 ^a^	8.4 ± 0.01 ^a^	5.6 ±0.25 ^b^	4.8 ±0.19 ^b^	4.7 ± 0.18 ^b^	4.7 ± 0.05 ^b^
TC2	7.8 ± 0.03 ^b^	9.1 ± 0.03 ^b^	9.0 ± 0.06 ^b^	8.9 ±0.09 ^b^	4.8 ± 0.14 ^c^	5.0 ± 0.14 ^c^	5.0 ± 0.17 ^c^	4.4 ± 0.17 ^c^
TC10	7.9 ± 0.04 ^b^	8.7 ± 0.09 ^b^	8.8 ± 0.09 ^b^	9.3 ±0.01 ^b^	4.7 ± 0.18 ^c^	4.8 ± 0.26 ^c^	4.4 ± 0.09 ^c^	4.4 ± 0.08 ^c^
TC16	7.7 ± 0.09 ^b^	9.2 ± 0.05 ^b^	9.0 ± 0.06 ^b^	9.4 ±0.05 ^b^	4.8 ± 0.12 ^c^	4.8 ± 0.24 ^c^	4.4 ± 0.07 ^c^	4.6 ± 0.09 ^c^
TC48	7.8 ± 0.14 ^b^	8.8 ± 0.09 ^b^	8.9 ± 0.11 ^b^	9.2 ±0.04 ^b^	4.5 ± 0.52 ^c^	4.2 ± 0.13 ^c^	3.8 ± 0.09 ^c^	4.1 ± 0.17 ^c^
TC50	7.8 ± 0.31 ^b^	9.1 ± 0.09 ^b^	9.2 ± 0.04 ^b^	9.3 ±0.10 ^b^	4.5 ± 0.12 ^c^	4.3 ± 0.16 ^c^	4.0 ± 0.12 ^c^	4.2 ± 0.05 ^c^
KACC91961P	7.6 ± 0.29 ^b^	8.9 ± 0.12 ^b^	9.0 ± 0.24 ^b^	9.0 ±0.08 ^b^	4.5 ± 0.19 ^c^	4.3 ± 0.42 ^c^	4.2 ± 0.14 ^c^	4.2 ± 0.07 ^c^

^1·^LAB: Lactic acid bacteria, CFU: Colony per unit, S- controls (Sterilized with Ethylene Oxide gas); Data were expressed as mean of standard deviation (Mean ± STD, *n* = 3); a,b indicates significant between control and LAB treated silages (*p* < 0.05): ^2·^Acidification of silages (pH); ^a^, ^b^, ^c^ Means with different letters within a column are significantly different at the 5% level.

**Table 2 microorganisms-07-00318-t002:** Antibacterial activity of silages extract fermented with TC48 and TC50 at different moistures levels.

Pathogenic Bacteria	Control Silage Extract				TC48 Inoculated Extract (*p* < 0.05) ^#^				TC50 Inoculated Extract (*p* < 0.05) ^#^			
	40%	50%	60%	70%	40%	50%	60%	70%	40%	50%	60%	70%
*E. faecalis*	3.7 ± 1.1	7.5 ± 0.8	10.5 ± 1.5	7.8 ± 1.2	13.5 ± 3.3	18.1 ± 3.3	22.0 ± 3.4	18.9 ± 1.3	10.3 ± 0.3	15.6 ± 0.9	27.6 ± 0.5	25.4 ± 0.3
*E. coli*	4.4 ± 1.0	8.2 ± 0.7	9.0 ± 1.0	8.2 ± 1.0	13.7 ± 2.4	18.3 ± 0.8	27.0 ± 1.7	24.0 ± 1.5	13.3 ± 0.1	19.6 ± 0.3	26.6 ± 0.5	23.6 ± 0.6
*P. aeruginosa*	2.4 ± 0.4	6.2 ± 1.1	5.5 ± 0.2	8.7 ± 1.6	12.8 ± 1.4	17.4 ± 1.4	26.0 ± 1.3	19.9 ± 0.8	9.8 ± 0.8	13.4 ± 0.4	20.0 ± 0.2	19.3 ± 1.2
*S. aureus*	4.2 ± 0.9	8.0 ± 0.5	7.5 ± 0.8	8.5 ± 0.5	12.9 ± 0.4	17.5 ± 0.4	27.6 ± 2.1	23.0 ± 2.9	8.5 ± 0.2	13.8 ± 0.2	26.6 ± 1.1	18.8 ± 3.0

Data were expressed as mean of standard deviation (mean ± STD, *n* = 3). Inhibitory zone was mentioned as millimeter (mm); ^#^
*p* < 0.05 significant between control silage and LAB treated silage extracts.

**Table 3 microorganisms-07-00318-t003:** TC48 and TC50 ferment different carbohydrate substrates—in vitro.

Name of Carbohydrates	TC48	TC50
Glycerol	0	0
Erythritol	+	++
d-Arabinose	0	+
l-Arabinose	0	+
d-Ribose	+	+++
d-Xylose	+	+++
l-Xylose	+++	+
d-Adonitol	+	+
Methyl-β-d-xiloside	+	0
d-Galactose	+	0
d-Glucose	+++	+++
d-Fructose	+++	+++
d-Mannose	+++	+++
l-Sorbose	+++	+++
l-Rhamnose	0	+++
Dulcitol	+	++
Inositol	0	+
d-Mannitol	0	++
d-Sorbitol	0	+++
Methyl- α-D-mannoside	0	+++
Methyl- α-D-glucoside	0	+
*N*-acetyl glucosamine	0	+++
Amygdalin	0	+++
Arbutin	+++	+++
Esculin ferric citrate	+++	+++
Salicin	+++	+++
d-Celiobiose	+++	+
d-Maltose	+++	+++
d-Lactose	+++	+++
d-Melibiose	0	+++
d-Trehalose	+	+
Inulin	+++	+++
d-Melezitose	0	0
d-Raffinose	0	+++
Amidon	+	0
Glycogen	+	0
Xylitol	+	0
Gentiobiose	+	+
d-Turanose	++	+
d-Lyxose	0	+++
d-Tagatose	0	+
d-Fucose	+++	+++
l-Arabitol	0	+
Potassium gluconate	0	0
Potassium 2-Ketogluconate	++	+

+++ (Strong fermentation); ++ (medium fermentation); + (low fermentation); 0 (no fermentation).

**Table 4 microorganisms-07-00318-t004:** Extracellular enzymes productions from TC48 and TC50 in vitro.

Enzymes	48	50
Control	0	0
Alkaline phosphatase	+	+++
Esterase (C_4_)	+	+++
Esterase lipase (C_8_)	++	++
Lipase (C_14_)	+	++
Leucine arylamidase	+++	+++
Valine arylamidase	++	+++
Cystine arylamidase	+	++
Trypsin-like serine protease	+	+
αChymotrypsin	+	+++
Acid phosphatase	++	+++
Naphthol-AS-biphosphohydrolase	+++	++
α-Galactosidase	+	+
β-Galactosidase	+++	+++
β-Glucuronidase	+	++
α-Glucosidase	+	+++
β-Glucosidase	++	++
*N*-Acetyl-β-glucosaminidase	0	0
α-Mannosidase	0	0
α-Fucosidase	0	+++

+++ (Strong production); ++ (medium production); + (low production); 0 (no fermentation).

**Table 5 microorganisms-07-00318-t005:** Antibiotic sensitivity patterns of TC48 and TC50—in vitro.

S. No.	Names of Antibiotics	Concentrations (μg)	TC48	TC50
1	Chloramphenicol (C)	50	SS	SS
2	Kanamycin (K)	30	SS	SS
3	Nitrofurantoin (NIT)	50	SS	SS
4	Tetracycline (TE)	100	SS	SS
5	Streptomycin (S)	25	SS	SS
6	Sulphafurazole (SF)	300	MS	MS
7	Colistin methane sulphonate (CL)	100	SS	SS
8	Dicloxacillin (D/C)	1	SS	MS
9	Ampicillin (AMP)	10	SS	SS
10	Amikacin (AK)	30	R	SS
11	Gentamicin (GEN)	10	SS	SS
12	Cefoxitin (CX)	30	R	SS
13	Cefalexin (CN)	30	SS	SS
14	Cefuroxime (CXM)	30	SS	SS
15	Co-Trimoxazole (COT)	25	MS	SS

<8 mm: Moderate susceptibility = MS; >10 mm: Strong susceptibility = SS: Resistant = R.

**Table 6 microorganisms-07-00318-t006:** Antibacterial activities of fresh and denatured ICS and ECS of TC48 and TC50 against pathogenic bacteria—well diffusion method.

**Bacterial Names**	**Fresh Intra Cellular and Extracellular Supernatants**			
	*E. faecalis*	*E. coli*	*P. aeruginosa*	*S. aureus*
TC48-ICS *	0	11.26 ± 3.60 ^a^	5.83 ± 1.25 ^b^	3.63 ± 1.18 ^b^
TC48-ECS ^#^	27.67 ± 2.52 ^a^	27.67 ± 2.08 ^a^	14.00 ± 2.65 ^c^	20.67 ± 4.04 ^b^
TC50-ICS *	0	0	5.66 ± 1.15	9.33 ± 1.52
TC50-ECS ^#^	27.67 ± 3.51 ^a^	26.67 ± 4.51 ^a^	28.33 ± 2.08 ^a^	21.67 ± 1.53 ^b^
**Bacterial Names**	**Denatured Intra Cellular and Extracellular supernatants**			
TC48-ICS *	0	0	0	0
TC48-ECS ^#^	18.77 ± 1.02 ^a^	10.90 ± 3.70 ^b^	7.07 ± 0.95 ^b^	11.77 ± 2.92 ^b^
TC50-ICS *	0	0	0	0
TC50-ECS ^#^	15.90 ± 1.65 ^a^	13.87 ± 4.51 ^b^	13.0 ± 1.48 ^b^	14.50 ± 1.53 ^b^

ECS: Extracellular supernatants; ICS: Intracellular supernatants; inhibitory zone was mentioned as millimeter (mm). Results were expressed as mean ± STD, *n* = 3), ^a, b^ and ^c^ indicates significant differences on LAB treatment between pathogenic bacteria; * ^#^
*p* < 0.05 significant differences between ICS and ECS of LAB on pathogenic bacteria.

**Table 7 microorganisms-07-00318-t007:** MIC and MBC ECS of TC48 and TC50 against different pathogenic bacteria.

Pathogenic bacteria	TC48-ECS (mg/mL)		TC50-ECS (mg/mL)	
	MIC	MBC	MIC	MBC
*E. faecalis*	5	10	5	10
*E. coli*	5	10	5	10
*P. aeruginosa*	10	20	5	10
*S. aureus*	10	20	10	20

ECS: Extracellular supernatants; MIC: Minimum inhibitory concentration; MBC: Minimum bactericidal concentration.

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
