# Peer review of "Probiotic and Triticale Silage Fermentation Potential of Pediococcus pentosaceus and Lactobacillus brevis and Their Impacts on Pathogenic Bacteria"

_microorganisms, 2019, doi:10.3390/microorganisms7090318_

Round 1

Reviewer 1 Report

To the authors,

The manuscript entitled " Probiotic and Triticale Silage Fermentation potential of Pediococcuspentosaceus and Lactobacillus  brevis and Their Impacts on pathogenic bacteria" describe the isolation of bacterial species from triticalesilage and the characterisation of selected isolates to determine their silage fermentation and probiotic capabilities.

The manuscript represents a significant amount of research, the outcomes of the project are clearly explained and the conclusions are sound. There are however a few suggestions to be considered before publication.

Seeing that the work presented in the manuscript is centered around the isolated LAB and the triticale silage, I would like the authors to provide more detail about the isolation procedures and silage treatment conditions described in sections 2.1 and 2.2 of the Materials and Methods.

Several sentences in the manuscript require rephrasing to adhere to proper English sentence structure. This was mainly limited to the abstract and introduction sections. Here are a few examples, but this is by no means a complete list and I would recommend using a professional editing service. Few examples from the abstract: Line 16: "For this purpose, one Pediococcus pentosaceus (TC48)
and Lactobacillus brevis (TC50) were isolated from fermented triticale silage." In summary, all of the tested new strains resulted in positive impacts isolated bacteria had a positive effect on at least one functional property of the silage during the fermentation process. Line 30: "However, the addition of P. pentosaceus (TC48) and L. brevis (TC50) could be indicated as the better for yielded the greatest silage quality improvement, having both with high antibacterial and probiotic properties." Line 19: give the full name of abbreviations at first use eg. ECS Line 37: just use "cattle" instead of "cattle animals" throughout the manuscript. Line 52: "grasses" Line 54: "growth" Line 56: "breed crop species" - did you mean hybrid species? Line 56: "German" Line 61: "It could replace the use of oats, ...." Line 62: "biomass, fast growth in Spring and ..." Line 78: "prepared samples with different moisture contents ..."` Line 90: "A hundred grams of each sample, at different moisture levels, were taken in triplicate and placed in vacuum ....."

Line 137: Please provide the strain acquisition numbers of all pathogens obtained from the culture collection. Line 206: " TC48 and, TC50 displayed adequate growth and ..." Line 240: Please provide a guide in the main text to indicate what "strong", "medium" and "low" represents for all the tables. Line 264: "..it demonstrated that something other than proteins .." Line 278: "Table 7" The quality of figure 3, 4, 6E and 6F should be improved. Line 400: reference should not be in italics Line 404: "disruption, So it's ..." ? Line 411: "survival" Line 415: ".. TC50 were survived... Line 423: "because of these strains get more.." Line 438: "climbing" - should this be clumping? Line 342: remove round bracket Line 347: growth Line 349: "goal was to make"'

There are many formatting issues involving double spaces of the absence of spaces throughout the manuscript, eg. Line 23: E. coli instead of E.coli and Line 22: supernatant(ECS) instead of supernatant (ECS).

The reference section does not have consistent formatting.

Author Response

 Subject: Revision of Manuscript ID: 578960

Title: Probiotic and Triticale Silage Fermentation potential of Pediococcus pentosaceus and Lactobacillus brevis and Their Impacts on pathogenic bacteria

Dear Editor in Cheif

Thank you very much for reviewers and editor comments of our above research paper and for allowing us to revise our above manuscript to amend the same for further consideration in your esteemed journal. The comments by reviewers about research paper make a quality manuscript. We have strengthened the introduction part of manuscript by including additional details and reformatted the silage production protocol. We carefully considered reviewers comments that colored the revisions made in the manuscript. Language of the manuscript has been checked carefully by the Harrisco Scientific English Research paper editing service. The paper has been modified accordingly with relevant changes appended separately. We do hope that the revised manuscript is now suitable for publication in Microorganisms

We look forward to your kind consideration.

Thank you

Sincerely Yours,

Ki Choon Choi

National Institute of Animal Science.

Cheonan.Republic of Korea-330-801.

Email: choiwh@korea.kr

Tel.: +82-41-580-6752.

Fax: +82-41-580-6779.

At the outset we thank the reviewers for their critical and judicious evaluation of our manuscript, and providing constructive suggestions for improving the quality and presentation of the manuscript. We have carefully considered the comments of the reviewers and revised the manuscript thoroughly taking all the points into consideration. Point wise response to the reviewer's comments is given below.

Comments from Reviewer 1

The manuscript entitled " Probiotic and Triticale Silage Fermentation potential of Pediococcuspentosaceus and Lactobacillus  brevis and Their Impacts on pathogenic bacteria" describe the isolation of bacterial species from triticalesilage and the characterisation of selected isolates to determine their silage fermentation and probiotic capabilities.The manuscript represents a significant amount of research, the outcomes of the project are clearly explained and the conclusions are sound. There are however a few suggestions to be considered before publication.

ANS. Thank you for your positive comments about our research paper, really, it encouraged us to make a good quality research paper in the field of agriculture microbiology in future.  

Seeing that the work presented in the manuscript is centered on the isolated LAB and the triticale silage, I would like the authors to provide more detail about the isolation procedures and silage treatment conditions described in sections 2.1 and 2.2 of the Materials and Methods.

ANS. Thank you for your comments, as per reviewer suggestion we have included a detailed protocol for isolation of lactic acid bacteria (Line No:85-93). Also, we have reframed silage production protocol in detail. We hope that these details could be sufficient to understand easily by the readers (Line Nos: 93-117)

Several sentences in the manuscript require rephrasing to adhere to proper English sentence structure. This was mainly limited to the abstract and introduction sections. Here are a few examples, but this is by no means a complete list and I would recommend using a professional editing service. Few examples from the abstract: Line 16: "For this purpose, one Pediococcus pentosaceus (TC48)and Lactobacillus brevis (TC50) were isolated from fermented triticale silage." In summary, all of the tested new strains resulted in positive impacts isolated bacteria had a positive effect on at least one functional property of the silage during the fermentation process. Line 30: "However, the addition of P. pentosaceus (TC48) and L. brevis (TC50) could be indicated as the better for yielded the greatest silage quality improvement, having both with high antibacterial and probiotic properties." Line 19: give the full name of abbreviations at first use eg. ECS Line 37: just use "cattle" instead of "cattle animals" throughout the manuscript. Line 52: "grasses" Line 54: "growth" Line 56: "breed crop species" - did you mean hybrid species? Line 56: "German" Line 61: "It could replace the use of oats, ...." Line 62: "biomass, fast growth in Spring and ..." Line 78: "prepared samples with different moisture contents ..."` Line 90: "A hundred grams of each sample, at different moisture levels, were taken in triplicate and placed in vacuum ....."

ANS: Thank you for your valuable comments. We would like to ask an apology for these careless mistakes and errors. We carefully checked the whole manuscript and took necessary changes in the manuscript. All changes were made in red color.

Line 137: Please provide the strain acquisition numbers of all pathogens obtained from the culture collection. Line 206: " TC48 and, TC50 displayed adequate growth and ..." Line 240: Please provide a guide in the main text to indicate what "strong", "medium" and "low" represents for all the tables. Line 264: "..it demonstrated that something other than proteins .." Line 278: "Table 7" The quality of figure 3, 4, 6E and 6F should be improved. Line 400: reference should not be in italics Line 404: "disruption, So it's ..." ? Line 411: "survival" Line 415: ".. TC50 were survived... Line 423: "because of these strains get more.." Line 438: "climbing" - should this be clumping? Line 342: remove round bracket Line 347: growth Line 349: "goal was to make"'  

ANS :Details of strong, medium and low given in the manuscript which indicates fermentation capability of carbohydrates by the isolated strains (Line Nos: 273-275)We have reframed sentences as per reviewer suggestion. All changes were made with red colo

-High-quality figures were given for figures 3, 4, 6E and 6F

 Asper reviewer suggestion we have given acquisition numbers for all pathogens (Line Nos: 162- 164)

There are many formatting issues involving double spaces of the absence of spaces throughout the manuscript, eg. Line 23: E. coli instead of E.coli and Line 22: supernatant (ECS) instead of supernatant (ECS).ANS: Yes we agreed with reviewer comments. We have reduced space between the words

The reference section does not have consistent formatting.

ANS: Thank you for your valuable suggestion. Here we used endnote software for reference preparation. However, we have reformatted all reference according to the journal guidelines.

Reviewer 2 Report

The paper  “Probiotic and triticale silage fermentation potential of Pediococcus pentosaceus…” addresses a topic worthy of investigation; the results could be of concern, but the paper does not deserve publication in its present form. There are important issues to be addressed.

MAJOR ISSUES

Materials and Methods

Generally, this section is not clear. After reading the results, I realized that the authors isolated some bacteria, performed an initial screening and selected the isolates  48 and 50. These isolates were identified through genotyping, used for fermentation and to simulate the transit into the gut. I think that the whole section should be revised to make clear this approach.

Section 2.1: How many samples did authors collect? Were they representative?

Section 2.3.2: The authors determined LAB and fungi. This is not correct. During vegetable fermentation, other groups should be assessed (spore-forming bacteria, enterobacteria etc..). Did authors have an idea of their evolution throughout fermentation?

Section 2.7.2: During the transit into the gut, the oral phase is important, but it is missing in the protocol used by authors. Why?

Section 2.8: I know that hydrophobicity is a screening index used as a quick tool and as an indirect measure of bacterial adhesion. Did authors confirm their data at least by biofilm formation?

Results

Table 1. Please use log-values and add (for each sample) standard deviation

Table 2. It is not correct to evaluate standard deviation amongst different samples. It should be assessed amongst the dependent and independent replicates of each sample. Revise accordingly!

Table 6. A multivariate analysis should help understanding the most important differences

Figure 6A and B. Please add the actual log values at the beginning and after the simulation or at least the difference (as log values). The use of survival ratio makes not clear the actual viability loss after the simulation of the transit into the gut.

MINOR ISSUEs

l. 48. For probiotic definition, use the new one reported by Hill et al. (2014)

l. 63-66: Please expand and define better the main goal of the paper and the specific objectives

l. 137. Please add the Collection Code of the pathogens

l. 194-197. Specify the post-hoc test run on the results after ANOVA

Author Response

Subject: Revision of Manuscript ID: 578960

Title: Probiotic and Triticale Silage Fermentation potential of Pediococcus pentosaceus and Lactobacillus brevis and Their Impacts on pathogenic bacteria

Dear Editor in Cheif

Thank you very much for reviewers and editor comments of our above research paper and for allowing us to revise our above manuscript to amend the same for further consideration in your esteemed journal. The comments by reviewers about research paper make a quality manuscript. We have strengthened the introduction part of the manuscript by including additional details, reformatted the silage production protocol, and mentioned the clear goal of this research paper. We carefully considered reviewers comments that colored the revisions made in the manuscript. Language of the manuscript has been checked carefully by the Harrisco Scientific English Research paper editing service. The paper has been modified accordingly with relevant changes appended separately. We do hope that the revised manuscript is now suitable for publication in MicroorganismsWe look forward to your kind consideration.

Thank you

Sincerely Yours,

Ki Choon Choi

National Institute of Animal Science.

Cheonan.Republic of Korea-330-801.

Email: choiwh@korea.kr

Tel.: +82-41-580-6752.

Fax: +82-41-580-6779.

At the outset, we thank the reviewers for their critical and accurate evaluation of our manuscript and providing constructive suggestions for improving the quality and presentation of the paper. We have carefully considered the comments of the reviewers and revised the manuscript thoroughly considering all the points. Point wise response to the reviewer's comments is given below

Comments from Reviewer 2

Materials and Methods

Generally, this section is not clear. After reading the results, I realized that the authors isolated some bacteria, performed an initial screening and selected the isolates  48 and 50. These isolates were identified through genotyping, used for fermentation and to simulate the transit into the gut. I think that the whole section should be revised to make clear this approach.

ANS: Yes, we agreed with reviewer comments, we have slightly reframed methods and materials as per reviewer suggestion for easy understanding by the readers. Our main aim is to isolate potent probiotic lactic acid bacteria for silage production. Our strong criteria for selection of LAB are; fast acidification of silage must have a strong antibacterial activity and significant probiotic potential. We isolated several lactic acid bacteria (LAB) from fermented triticale sample. Fifty-eight LAB exhibits significant antibacterial activity against E.coli and S. aureus. Later we have selected two LAB based on its acidification efficiency and organic acid production. These two LAB further subjected into biochemical and molecular characterization, antibacterial activity against five pathogenic bacteria and probiotic potential were also studied

Section 2.1: How many samples did authors collect? Were they representative?

ANS: Triticale grass was collected at different places in the same land from Jangsoo, South Korea. The samples were brought aseptically to the laboratory for isolation of lactic acid bacteria (LAB). One gram of the sample was dissolved in 10 ml distilled water and kept in a rotary shaker at 150rpm for 30min. After, tenfold serial dilution was performed. Serially diluted samples were spread on Man Rogosa and Sharpe agar medium (MRS agar; CONDA, Madrid, Spain). Plates were incubated at 30°C for 72h, observed bacteria growth and then purified on MRS agar. LAB colonies were further streaked on Bromo cresol purple agar medium (BCP), BCP a selective medium to confirm the identity of LAB strains. All strains were maintained on MRS agar medium at 4°C for short time storage

Section 2.3.2: The authors determined LAB and fungi. This is not correct. During vegetable fermentation, other groups should be assessed (spore-forming bacteria, enterobacteria etc...). Did authors have an idea of their evolution throughout fermentation?

ANS: Thank you for your kind comments, silage production was performed by ensiling method. It has more attention because it provides consistent, reliable and predictable feed supply for animal productions. Addition of LAB during silage production prevents undesirable microbial growth by reducing the pH of the surrounding environment via producing a higher amount of lactic acid. Successful fermentation indicates higher amounts of lactic acid and lowered pH of silage with good smells. If a failure in fermentation indicates (higher clostridia fermentation) poor lactic acid production and higher butyric acid content with bad smells. So, the addition of LAB to silage overcomes these issues by preventing undesirable microbial growth by its competence growth between LAB and undesirable bacteria for long time storage also. Selecting and adding potent LAB is a key factor for silage production

Section 2.7.2: During the transit into the gut, the oral phase is important, but it is missing in the protocol used by authors. Why?

ANS: Yes, we agreed with reviewer comment, in the current study, we did not analyze oral tolerant test for selected LAB. In general, we analyze survival ability in simulated gastric juice, duodenal juice and intestinal juice. Most of the researchers worldwide follow the same protocol for potent probiotic selection. However, we can utilize your suggestion in our future study.    

Section 2.8: I know that hydrophobicity is a screening index used as a quick tool and as an indirect measure of bacterial adhesion. Did authors confirm their data at least by biofilm formation?

ANS: Thanks for asking this question. We never performed biofilm formation assay for these LAB. In future studies, we can include your valuable opinion. We have may plan to investigate the biological potential of these isolated strains.

Table 1. Please use log-values and add (for each sample) standard deviation

As per reviewer suggestion, we have included the results in log values and its standard deviation for each.  

Table 2. It is not correct to evaluate standard deviation amongst different samples. It should be assessed amongst the dependent and independent replicates of each sample. Revise accordingly!

ANS: We agreed with reviewer comments, as per reviewer suggestion, we have included standard deviation for each.  

Table 6. A multivariate analysis should help understanding the most important differences

ANS: Thanks, we performed multivariate and One-way ANOVAs also which includes post-hoc, Duncan test and descriptive parameter. We compared effects between ICS and ECS on pathogenic bacteria, ICS and ECS effects between different pathogenic bacteria

Figure 6A and B. Please add the actual log values at the beginning and after the simulation or at least the difference (as log values). The use of survival ratio makes not clear the actual viability loss after the simulation of the transit into the gut.

ANS: Thank you for the valuable suggestion, actually 6A, B and C indicates survival ability in harsh condition of GI tract. These conditions include low pH, bile salt environment and higher pH. So we mentioned the percentage of LAB survival under these conditions. It provides a clear picture between 0hour and after 3hours. We hope that the reviewer can understand this explanation.

For probiotic definition, use the new one reported by Hill et al. (2014)

ANS: Thanks, we have include this reference in introduction part (line No:53- 55)

l. 63-66: Please expand and define better the main goal of the paper and the specific objectives

Thanks, the purpose of the present study was to identify and characterize efficient probiotic lactic acid bacteria (LAB) and investigate their effects on triticale silage fermentation, and cattle pathogenic bacterial growth

l. 137. Please add the Collection Code of the pathogens

ANS: Thanks, we have included these details in method section (Line No 161-165)

l. 194-197. Specify the post-hoc test run on the results after ANOVA

Many thanks; we always performed statistical analysis for all samples by Oneway ANOVA and multivariate analysis with posthoc test, Duncan test and descriptive analysis with 0.05 level significant.

All numerical data were obtained from three independent experiments. Results are presented as means ± STD Significant differences were analyzed with SPSS16 software using analysis of variance (one-way ANOVA, multivariate analysis, included post hoc, Duncan and descriptive analysis parameters) followed by the least significance difference test. P values of less than 0.05 were considered as statistically significant

Round 2

Reviewer 2 Report

The authors replied to my issues; I have some concerns concerning the growth of clostridia, enterobacteria. I think that the authors should address it and maybe adding in the conclusions that further investigations are required for this challenge

Author Response

Reviewer 1

Subject: Revision of Manuscript ID: 578960

Title: Probiotic and Triticale Silage Fermentation potential of Pediococcus pentosaceus and Lactobacillus brevis and Their Impacts on pathogenic bacteria

Dear Editor in Cheif

Thank you very much for reviewer and editor comments of our above research paper and for allowing us to revise our above manuscript to amend the same for further consideration in your esteemed journal. We carefully considered reviewers comments that colored the revisions made in the manuscript. The paper has been modified accordingly with relevant changes appended separately. We do hope that the revised manuscript is now suitable for publication in Microorganisms

We look forward to your kind consideration.

Thank you

Sincerely Yours,

Ki Choon Choi

National Institute of Animal Science.

Cheonan.Republic of Korea-330-801.

Email: choiwh@korea.kr

Tel.: +82-41-580-6752.

Fax: +82-41-580-6779.

Reviewer comment

The authors replied to my issues; I have some concerns concerning the growth of clostridia, enterobacteria. I think that the authors should address it and maybe adding in the conclusions that further investigations are required for this challenge

ANS: Thanks many, As per reviewer suggestion, we have modified conclusion slightly as this result obtained at the laboratory level experiment and must be needed to confirm clostridial, and enterobacteria growth before/after inoculants treatment in experimental silages under field conditions (Line No 513-515).

Also, we have included graphical abstract for easy understanding of experimental protocol by the readers.